# Patch-walking, a coordinated multi-pipette patch clamp for efficiently finding synaptic connections

**Mighten C Yip[1]\*, Mercedes M Gonzalez[1], Colby F Lewallen[2], Corey R Landry[3], Ilya Kolb[4], Bo Yang[1], William M Stoy[5], Ming-fai Fong[3], Matthew JM Rowan[6], Edward S Boyden[7,8,9], Craig R Forest[1]\***

[1]George W Woodruff School of Mechanical Engineering, Georgia Institute of Technology, Atlanta, United States; [2]Ocular and Stem Cell Translational Research Section, Ophthalmic Genetics and Visual Function Branch, National Eye Institute, National Institute of Health, Bethesda, United States; [3]Wallace H. Coulter Department of Biomedical Engineering, Georgia Institute of Technology, Atlanta, United States; [4]GENIE Project Team, Janelia Research Campus Howard Hughes Medical Institute, Ashburn, United States; [5]Department of Electrical Engineering, Columbia University, New York, United States; [6]Department of Cell Biology, Emory University, Atlanta, United States; [7]Department of Brain and Cognitive Science, Massachusetts Institute of Technology, Cambridge, United States; [8]McGovern Institute for Brain Research, Massachusetts Institute of Technology, Cambridge, United States; [9]Howard Hughes Medical Institute, Cambridge, United States

**\*For correspondence:**
mighten.yip@gmail.com (MCY);
cforest@gatech.edu (CRF)

## eLife Assessment

This technical study presents a novel sampling strategy for detecting synaptic coupling between neurons from dual pipette patch-clamp recordings in acute slices of mammalian brain tissue in vitro. The authors present **solid** evidence that this strategy, which incorporates automated patch clamp electrode positioning and cleaning for reuse with strategic neuron targeting, has the potential to substantially improve the efficiency of neuronal sampling with paired recordings. This technique and the extensions discussed will be **useful** for neuroscientists wanting to apply or already conducting automated multi-pipette patch clamp recording electrophysiology experiments in vitro for neuron connectivity analyses.

**Abstract** Significant technical challenges exist when measuring synaptic connections between neurons in living brain tissue. The patch clamping technique, when used to probe for synaptic connections, is manually laborious and time-consuming. To improve its efficiency, we pursued another approach: instead of retracting all patch clamping electrodes after each recording attempt, we cleaned just one of them and reused it to obtain another recording while maintaining the others. With one new patch clamp recording attempt, many new connections can be probed. By placing one pipette in front of the others in this way, one can 'walk' across the mouse brain slice, termed 'patch-walking.' We performed 136 patch clamp attempts for two pipettes, achieving 71 successful whole cell recordings (52.2%). Of these, we probed 29 pairs (i.e. 58 bidirectional probed connections) averaging 91 μm intersomatic distance, finding three connections. Patch-walking yields 80–92% more probed connections, for experiments with 10–100 cells than the traditional synaptic connection searching method.

## Introduction

To elucidate the mechanisms that regulate memory formation, perception, decision-making, and other cognitive functions, scientists seek to measure brain activity at the resolution of individual neurons (*Segev et al., 2016*). However, neurons do not operate in isolation; cognitive function relies on chemical communication between these individual cells, giving rise to neural networks across the entire brain. These connections, or synapses, transmit information and measuring their strength, direction, and other properties is essential to unraveling how the brain works. Yet, there are significant challenges that make finding, measuring, and comprehending these dynamic synaptic connections time consuming and low throughput.

Patch clamp recording remains the gold-standard technique for high-fidelity electrophysiological measurements for studying individual neurons and their synaptic connections in the living brain (*Sakmann and Neher, 1984*; *Stuart et al., 1993*; *Markram et al., 1997*). Patch clamp, with subthreshold resolution and millisecond precision, has enabled studies ranging from mapping the healthy rodent brain to characterizing the behavior of single cells in neurodegenerative diseases (*van den Hurk et al., 2018*; *Castañeda-Castellanos et al., 2006*; *Wang et al., 2015*). However, in return for superior signal quality as compared with other methods (*Chen et al., 2022*; *Meyer et al., 2018*), the traditional patch clamp technique remains low throughput since it is manually laborious and time-consuming (*Hamill et al., 1981*).

Whole cell patch clamp recordings require forming an electrical connection between the recording electrode and the membrane of an individual neuron. This electrical connection requires a high resistance seal between the neuron's membrane and a clean microelectrode pipette. Detaching the pipette from this seal leaves behind residual membrane material that inhibits the formation of a new connection with another neuron. Thus, each pipette must be cleaned or replaced after each recording. When studying cells in intact tissue such as brain slices, skill is needed to avoid neighboring cells and account for tissue deformation. Even skillful efforts only yield around 10 cells recorded per day, and 'whole cell' success rates are highly variable, typically ranging from 30% to 90%of attempts resulting in successful patch clamp recordings, even for experienced users (*Campagnola et al., 2022*; *Wood et al., 2004*; *Hamill et al., 1981*).

Scaling traditional, manual patch clamp apparatus to multiple pipettes, in order to obtain synaptic connections between cells, has therefore been extraordinarily technically challenging. The efforts of, for example Perin (*Perin et al., 2011*) and Tolias (*Jiang et al., 2015*) are laudable, but can require years of effort to overcome low throughput and yield.

Recently, patch clamp recording efficiency and throughput has increased due to improvements in automated pressure control systems, new algorithms for automated pipette movements guided by visual or electrical signals, and pipette cleaning, rather than reuse (*Wu et al., 2016*; *Kodandaramaiah et al., 2018*; *Kodandaramaiah et al., 2012*; *Koos et al., 2021*; *Stoy et al., 2017*; *Harrison et al., 2015*; *Holst et al., 2019*). In this way, we have previously developed a robotic system, 'the PatcherBot', capable of performing unattended, multi-hour patch clamp experiments in brain slices, with a whole cell success rate of 51%(*Kolb et al., 2019*). These advances rely on the concurrent discovery that pipettes can be reused, rather than replaced after each recording attempt (*Kolb et al., 2016*; *Landry et al., 2021*; *Yip, 2023*). These improvements have enabled novel drug screening assays (*Perszyk et al., 2021*), deep in-vivo recordings (*Stoy et al., 2017*), voltage indicator screening, and fluorescent cell targeted patch clamp (*Wu et al., 2016*).

In the field of connectomics and synaptic physiology, several groups have developed methods for obtaining semi-automated patch clamp recordings of synaptically connected neurons (*Wang et al., 2015*; *Peng et al., 2019*; *Perin and Markram, 2013*; *Kodandaramaiah et al., 2018*; *Campagnola et al., 2022*). In the most impressive examples, large-scale connectomics studies have recently emerged from the Allen Institute for Brain Science and Geiger lab. At the Allen Institute, 20,949 connections were probed in the mouse brain (*Campagnola et al., 2022*). The efficiency of this effort over 1700 experiments, on average, yielded around 12 potential connections probed per experiment. The Allen Institute leveraged an eight-pipette setup that successfully connected to an average of four neurons per recording, resulting in an average of 12 possible connections ($n^2 - n = 12$) per experiment.

Peng et al. used the pipette reuse method (*Kolb et al., 2016*) to increase the number of potential connections probed from 12 to 41 (approximately n=7 cells patched simultaneously) on a comparable

eight pipette apparatus (*Peng et al., 2019*). Notably, these papers from the Allen Institute and the Geiger lab used eight manipulators, currently obtainable by only a handful of labs due to complexity and cost.

In all previous efforts, both manual and automated, in which multiple pipettes (referred to as multi-patching) are used to probe for synaptic connections, the experimental approach has involved (1) obtaining as many simultaneous recordings as possible, (2) probing their connections, and then (3) retracting all pipettes.

Recognizing how much effort and skill is necessary to obtain many simultaneous recordings, coupled with the advantages of pipette reuse and automation, we hypothesized a novel approach. If instead of retracting all pipettes, perhaps just one of them could be cleaned and reused to obtain a new whole cell recording while maintaining the others. Thus, with one new patch clamp recording attempt, many new connections can be probed. By placing one pipette in front of the others in this way, one can 'walk' across the tissue, which we term 'patch-walking.' Thus, in this work, we introduce the theory, methods, and experimental results for a fully automated in vitro approach with a coordinated pipette route-planning to 'patch-walk' across a brain slice. We demonstrate efficiently recording dozens of neurons using a two-pipette apparatus for finding synaptic connections. Here, we show that this approach, as compared with the traditional approach, increases the rate of potential neurons probed, decreases experimental time, and enables sequential patching of additional neurons.

## Results
### Mathematical modeling

The total possible number of connections probed using the traditional method of synaptic patch clamp recording can be expressed as a function of number of recorded cells ($n$), and number of pipettes in the multi-patch apparatus ($p$), as

$$\text{possible connections}_{\text{traditional}} = \frac{n}{p}(p^2 - p). \tag{1}$$

Similarly, the total possible number of connections probed using the patch-walking method can be expressed as

$$\text{possible connections}_{\text{patch−walk}} = (p^2 - p) + 2(p - 1)(n - p). \tag{2}$$

To visualize the advantage of patch-walking over the traditional method, these two equations can be represented as a matrix of potential probed connections. For example, the total number of possible connections using the traditional method and patch-walking for a two pipette apparatus is depicted in *Figure 1A and B*, respectively. Using these equations, patch-walking is always preferable in practice for $n > p$. Furthermore, one can expect the improvement in number of connections probed to approach double as $n$ approaches infinity. For practical cases (apparatus with 2–8 pipettes), patch-walking yields 80–92% more probed connections, or efficiency, for experiments with 10–100 cells than the traditional synaptic connection searching method.

### Dual-patching experiment

We built the apparatus (*Figure 1C*) and developed the software (*Figure 1D*) to perform patch-walking with two manipulators. We first conducted a dual-patch throughput experiment for two pipettes patching in a brain slice without testing for connectivity. In 33 patching attempts (18 attempts for pipette 1 and 15 attempts for pipette 2), we achieved whole cell success rates for pipette 1 of 44.4% (n=8/18 successful whole cells) and pipette 2 of 46.7% (n=7/15 successful whole cells). This is similar to success rates for manual patching as well as previously reported automated patch clamp robots (43–51% for *Kolb et al., 2019*). This result demonstrates the expected throughput and yield of these independent, uncoordinated pipettes.

Next we implemented the coordinated, dual-patching robot. In this series of experiments, we performed patch clamp attempts on 136 cells from 7 animals over a corresponding 7 days, with 2–3 slices per animal. Out of 136 patch clamp attempts for both pipettes, we achieved 71 successful whole cell recordings (52.2%). This is again comparable to previously reported automated patch clamp work

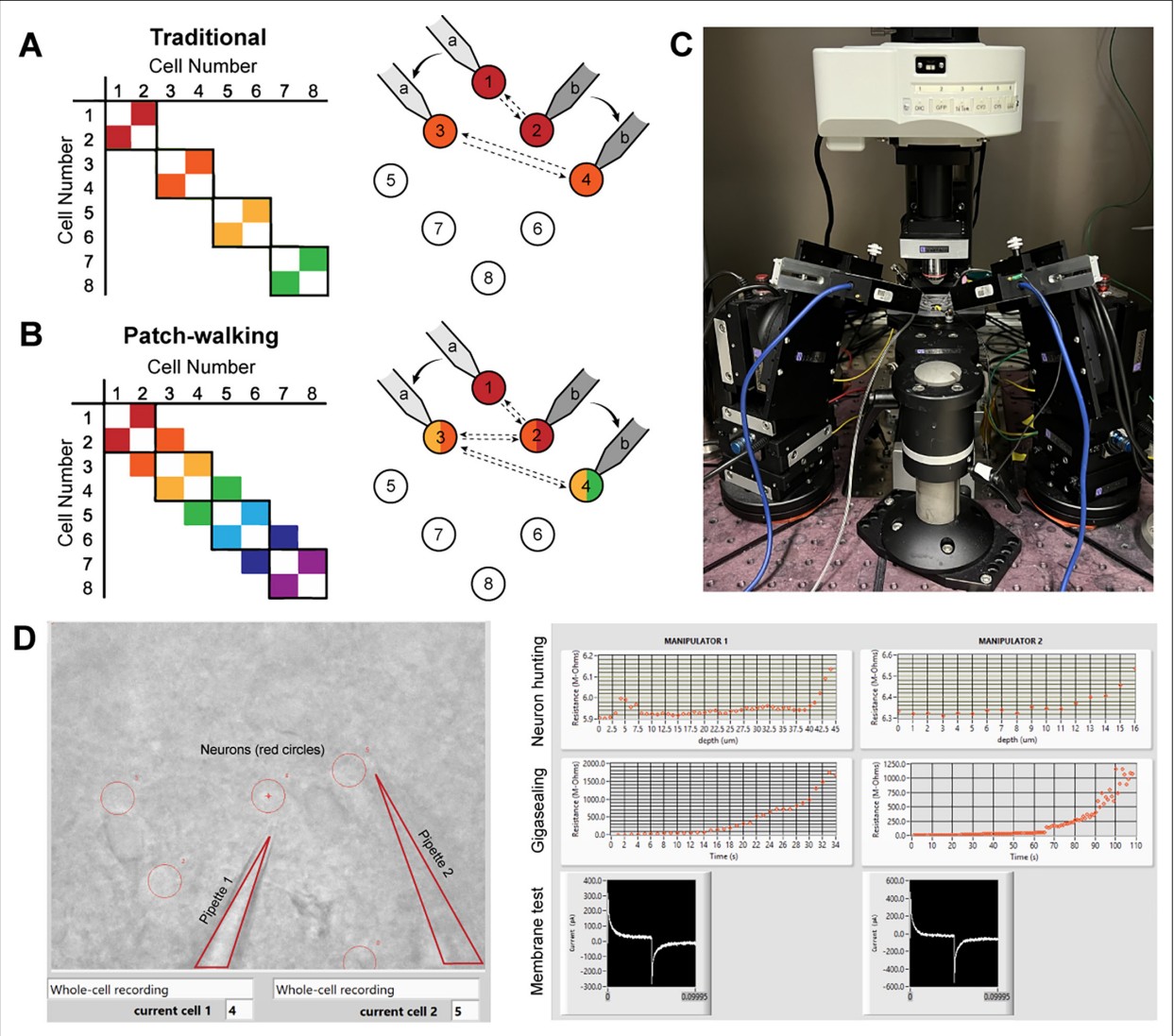

**Figure 1.** Patch-walk methodology and apparatus. (**A-B**) Schematically, for patch clamp apparatus with two pipettes in search of synaptically connected neurons to record; colored squares represent connections that can be probed using the traditional approach (**A**) as compared to patch-walking (**B**). In this schematic, n=8 cells were patched by $P$=2 pipettes, either in groups of two (**A**), which yields two possible connections, or by walking across the tissue (**B**), which yields almost double the number of possible connections. (**C**) A multi-patching apparatus with two pipettes was built with automated pressure control and manipulator movement. (**D**) The software interface used for patch-walking. On the left is the view of the brain slice under the microscope, with the two pipettes highlighted by triangles and user selected cell locations indicated by red circles. On the right are plots used to monitor each step of the patch clamp process: neuron hunting, gigasealing, and membrane test waveform (to monitor break-in state).

such as *Kolb et al., 2019* (51%), *Wu et al., 2016* (43.2%), and *Koos et al., 2021* (63.6% for rat, 37% for human). In addition, our success rates fall within the success range of manual users (30–80%) on differential interference contrast-based patch clamp systems (*Vera Gonzalez et al., 2023*). Thus, coordinating the motion of the pipette via the patch-walking algorithm does not deteriorate the success rate.

A representative brain slice with a box highlighting the experimental brain regions of interest is shown in *Figure 2A*. We patched in the somatosensory cortices as well as the primary visual cortex, primarily in L2/3, L4, and L5. In *Figure 2B* are histograms showing the distribution of the time it took to achieve a simultaneous recording (left, n=44), the intersomatic distance between neurons that were patch clamped simultaneously (center, n=44), and the time required to achieve gigaseal (time between increased resistance during neuron hunting step and achieving giga-ohm seal) for all cells (n=71). In *Figure 2C* are the distributions of whole cell properties (capacitance, tau, input resistance,

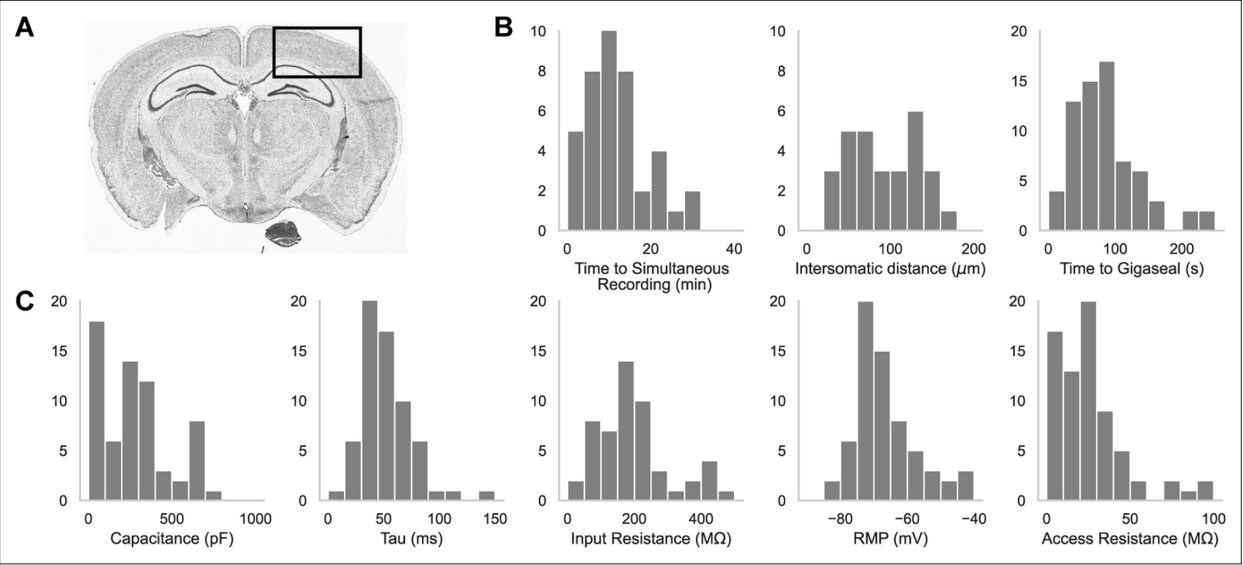

**Figure 2.** Dual-patching throughput and quality metrics. (**A**) An image of a brain slice with a box highlighting the brain region used for experiments: the somatosensory and visual cortices. (**B**) Histograms of patch clamp metrics: time to achieve simultaneous recording (n=44), distance between neurons during paired recordings (n=44), and amount of time to achieve gigaseal after a neuron is detected by the pipette (n=71), and (**C**) Membrane capacitance, time constant (tau), input resistance, resting membrane potential, and access resistance of all cells recorded during patch-walking experiments (n=71).

resting membrane potential, and access resistance) of all cells. We were able to achieve paired patch clamp recordings between two pipettes in an average of 12.6±7.5 min as the pipettes walked across the slice. The average distance between two neurons for screened for connections was 91.6 ± 0.2 µm. The cells in paired recordings were held in whole cell configuration up to 45 min.

We demonstrate a connectivity matrix similar to those done by previous labs such as Peng et al. and the Allen Institute for Brain Science (*Campagnola et al., 2022*; *Peng et al., 2019*). Of the 71 whole cell recordings we recorded from the robot, we report a yield of 44 paired recordings using our patch-walking technique. In comparison, if we had screened the same 71 neurons for connections using the traditional method, we would have screened for 71/2=35 paired recordings. Therefore, our patch-walking algorithm screened for 9 more possible paired recordings for the 71 neurons we tested. Of those 44 recordings, 29 paired recordings (i.e. 58 probed connections) passed quality control checks and were used to validate the efficiency of the patch-walking algorithm, resulting in 3 found synaptic connections. From *Equations 1; 2*, we see the additional connections screened with the effect of patch-walking.

According to *Perin et al., 2011*, at an intersomatic distance of 91.6 ± 0.171 µm, the expected connection probability is 16.9% for each paired recording. Assuming they are independent, we would expect greater than 50% probability of getting at least one connection after just three paired recordings. According to binomial probability theory, we had a probability of 89% to find 3 connections with 29 paired recordings.

*Figure 3* shows a connectivity matrix (as in *Figure 1B*), a spatial representation of the cells patched cells and connection probed, as well as a representative connection found between two cells. The matrix in *Figure 3A* shows the whole cell current clamp protocol described previously (black traces in leftmost column). During paired recordings, one cell would be stimulated in current clamp (traces along the diagonal). Recording color corresponds to a pair of cells tested for connectivity as in *Figure 1*, where each color has two traces because each pair of cells can be connected bidirectionally. The nomenclature for each row and column is $n.p$ where $n$ represents the cell number, in this case ranging from 1 to 7, since 7 total cells were patched between both pipettes, and $p$ labeled either a or b represents each of the two pipettes. Cell 7 exhibited signs of decreased cell health, likely due to the duration of the experiment and increasing physical disruptions to the slice during patch-walking. The representative connection shown in more detail in *Figure 3C* was found between cells 1 and 2,

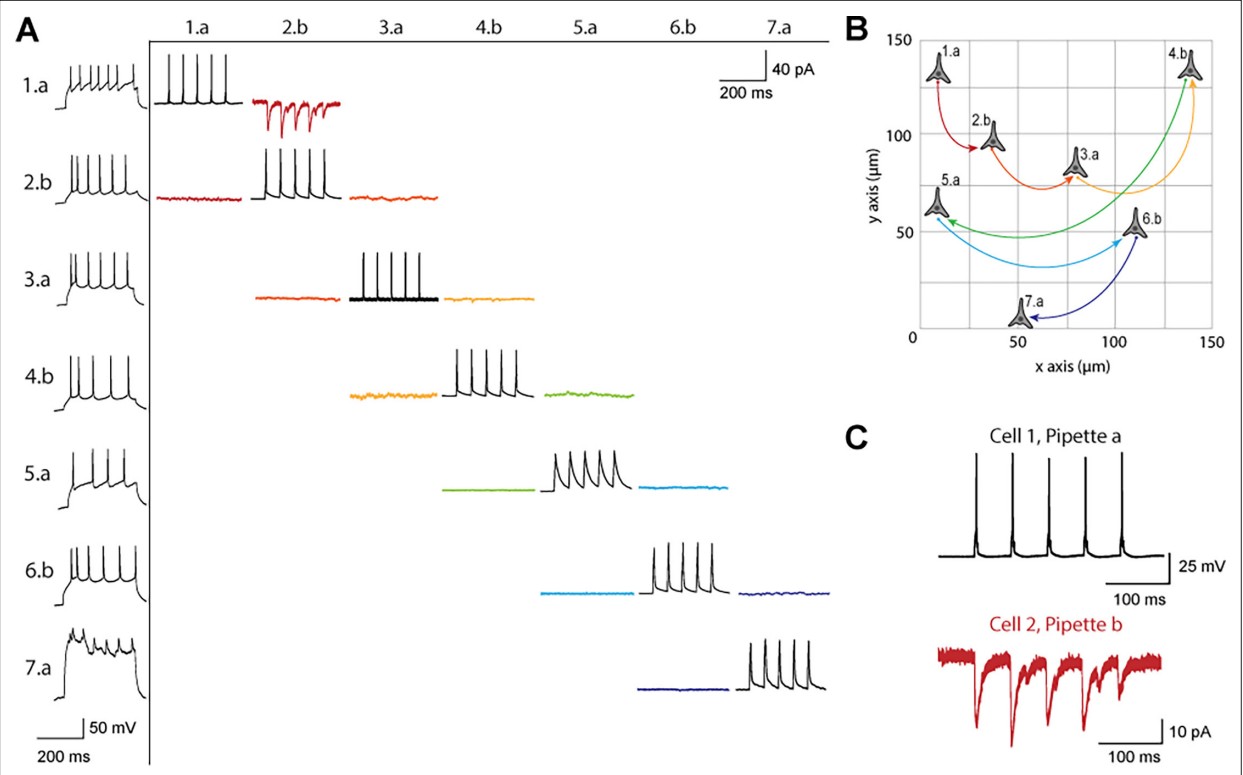

**Figure 3.** Connectivity matrix and recordings using patch-walking. (**A**) Matrix of voltage and current traces from seven neurons in one acute brain slice recorded using the patch-walking algorithm for the robot. Left column shows the firing pattern of the recorded neurons. Cells are numbered such that the number represents the cell and the letter represents the manipulator (a or b). Scale bars: Horizontal 200ms for firing pattern and connection screening. Vertical 40 mV for action potentials, 50 pA for postsynaptic traces. (**B**) Patch-walking scheme of all neurons from the experiment matrix in (**A**). The curved lines between neurons represent probed connections in the matrix in (**A**). (**C**) The probed connection from the connectivity matrix in (**A**). The stimulus was sent to cell 1 (black) and the response from cell 2 (red) was recorded and averaged over three sweeps.

with pre-synaptic cell 1 (black) stimulated and cell 2 (red) recording in voltage clamp the post synaptic currents elicited.

## Discussion

We introduce a variation on the multi-patching technique which we termed patch-walking. Patch-walking enables, theoretically, almost twice the number of connections to be probed on a patch clamping apparatus for a given number of cells patched and pipettes on the rig. We recommend this method for those searching for local synaptic connections using an apparatus with a small number (e.g. two) of pipettes, such as we have shown. Additionally, patch-walking causes less tissue damage because it only requires one new pipette to enter the slice for each connection being probed, compared to the traditional method that needs two new pipettes for each connection. Further, this method saves time between probed connections since only one pipette is moved at a time and it enables more recordings from a tissue before cell death which is advantageous for studying rare tissues such as human brain samples.

For scaling patch-walking beyond the apparatus described here, we caution that one must take into account pipette collisions and choose cell-pipette assignments carefully. Future work to improve upon patch-walking could include developing an optimal route-planning path such as the Monte Carlo Tree algorithm as a strategy to optimize the pipette-cell assignments, considering (1) spacing pipettes apart in order to avoid collisions between the fragile glass pipettes while (2) maximizing the probability of a connection. Specifically, we would recommend a threshold of inter-cell distance to be less than 200 µm in order to have at least a 10%probability of connection according to *Perin et al., 2011*.

The patch-walking algorithm can make multiple-pipette patch-clamp electrophysiology more accessible to a wider range of laboratories that usually conduct simultaneous recordings with several manipulators. For example, conducted studies such as Galarreta et al. studying a network of parvalbumin fast-spiking GABAergic interneurons could use this technology to make them more efficient to days worth of experiments rather than months (*Galarreta and Hestrin, 2002*). Further, the robot could also be altered to include users in the loop if they want to have control over certain aspects of the patching process or enable experienced patchers with digital pressure control. Even the best human electrophysiologists can only control one manipulator at a time, but the robot can control multiple pipettes, pressure regulators, and command signals independently. Patch-walking offers throughput improvements over manual patching, especially for those looking to utilize paired recordings in their experiments (*Bartos et al., 2001*; *Grosser et al., 2021*; *Linders et al., 2022*; *Qi et al., 2015*).

Out of the 29 paired recordings, we found 3 synaptic connections. While this number of connections is lower than predicted according to *Perin and Markram, 2013* based on the intersomatic distances between these cells, we hypothesize that this is most likely due to biological variation.

While the patch-walking method provides an efficient means to probe for synaptic connections using two pipettes, introducing additional pipettes presents notable challenges. Specifically, as pipettes maneuver into and out of the brain slice, tissue deformation often disturbs any pipettes in whole cell configuration.

Future applications and variations in patch-walking could include the use of channelrhodopsin-assisted circuit mapping *Abdelfattah et al., 2023*; *Petreanu et al., 2007* to enable larger circuit mapping with multiple patch electrodes. Patch-walking could also be used for fluorescent-targeted cells wherein one pipette could target a specific subset of cells while the other pipette would probe off-target cells. A third alternative could be that one pipette patches a deep cell and stays patched onto it while secondary pipettes continue to automatically patch other cells and search for connections. Additionally, this patch-walk protocol could also be implemented into manual recording approaches, leveraging the idea that only one pipette has to patch onto a new cell to test for connections, as opposed to two pipettes. Future work can include morphological identification or layer-to-layer connectivity studies. Further, machine learning algorithms to detect specific neuronal subtypes could be integrated for improved, real-time route-planning (*Yip et al., 2021*). From the presented methodology of patch-walking and potential future applications, patch-walking can be a useful tool to study synaptic connectivity, especially for researchers new to the field of single-cell electrophysiology.

## Methods
### Automated patch clamp apparatus

We designed and implemented an experimental apparatus to demonstrate the utility of patch-walking. The apparatus features a standard electrophysiology rig with two PatchStar micromanipulators. Samples (mouse brain slices) were imaged using a 40 X objective (LUMPLFL40XW/IR, NA 0.8, Olympus) on a motorized focus drive, illuminated under differential interference contrast microscopy (DIC) with an infrared light-emitting diode (780 nm), and captured with a Rolera Bolt camera (QImaging). We used a peristaltic pump (120 S/DV, Watson-Marlow) to perfuse the brain slices with buffer solution. We utilized the brain slice sample holder with integrated cleaning and rinse solution chambers as described previously (*Kolb et al., 2016*). We followed the cleaning protocol as suggested by *Kolb et al., 2016*, however we did not include rinsing in the cleaning protocol because recent literature found that there is no impediment to the whole cell yield or quality of recording (*Landry et al., 2021*; *Peng et al., 2019*).

Electrode pressure was controlled using a custom pipette pressure controller enabled up to four-channels, adapted from prior work (*Kolb et al., 2019*). Briefly, for each pipette, pressure was controlled by a±10 psi regulator (ProportionAir) using an analog (0–10 V) control signal. The control signal for each regulator was generated by a microcontroller (Arduino Due) via a digital-to-analog converter (MAX539, Maxim Integrated). In order to minimize valve switching to efficiently scale up the patcherBot to multiple manipulators and pipette pressure control, a custom printed circuit board was developed to control up to a maximum of four manipulators. Individual pressure regulators for each pipette were necessary to ensure that different pressures could be maintained on each pipette. The custom pressure controller regulates house-air line to deliver –500 to +700 mbar (relative to sea-level)

using an inline venturi tube (SMC) and solenoid valve (Parker Hannifin) for rapid pressure switching (*Kodandaramaiah et al., 2016*; *Kolb et al., 2019*; *Kodandaramaiah et al., 2012*; *Perszyk et al., 2021*).

For real-time electrophsyiology feedback and collection, we used the Multiclamp 700B amplifier (Molecular Devices), cDAQ-9263/9201 and USB- 6221 OEM data acquisition boards (National Instruments), and Axon Digidata 1550B. The two data acquisition boards were each assigned to a manipulator in order to simultaneously acquire different signals from each pipette. This is particularly important for asynchronous and independent pipette control. Machine vision-based pipette calibration and position correction was performed according to *Gonzalez et al., 2021* to correct for small micromanipulator position errors.

Following brain slice preparation and pipette fabrication, filling, and installation, pipette location was calibrated according to *Kolb et al., 2019* for all pipettes, resulting in a 'home' position as described for each pipette.

## Brain slice preparation

All animal procedures were in accordance with the US National Institutes of Health Guide for the Care and Use of Laboratory Animals and were approved by the Institutional Animal Care and Use Committee at the Georgia Institute of Technology (A100359). For the brain slice experiments, male mice (C57BL/6, P19–P36, Charles River) were anesthetized with isofluorane, and the brain was quickly removed. Coronal sections (300 μm thick) were then sliced on a vibratome (Leica Biosystems VT1200S) while the brain was submerged in ice-cold sucrose solution containing (in mM) 40 NaCl, 4 KCl, 1.25 $NaH_2PO_4 \cdot H_2O$, 7 $MgCl_2$, 25 $NaHCO_3$, 10 D-Gluocse, 0.5 $CaCl_2 \cdot 2H_2O$, 150 Sucrose (pH 7.3–7.4, 300–310 mOsm). The slices were incubated at 37 °C for 1 hr in neuronal artificial cerebro-spinal fluid (aCSF) consisting of (in mM) 124 NaCl, 2.5 KCl, 1.25 $NaH_2PO_4 \cdot H_2O$, 1.3 $MgCl_2$, 26 $NaHCO_3$, 10 D-Glucose, 2 $CaCl_2 \cdot 2H_2O$, 1 L-Ascorbate$\cdot H_2O$ (pH 7.3–7.4, 290–300 mOsm). Prior to recording, the slices were maintained at room temperature for at least 15 min (22–25 °C). The sucrose solution and neuronal ACSF were bubbled with 95%$O_2$/5%$CO_2$. Recordings were performed in mouse primary visual area and somatosensory cortex.

## Patch-clamp recording

Borosilicate pipettes were pulled on the day of the experiment using a horizontal puller (P-1000, Sutter Instruments) to a resistance of 4–6 MΩ. The intracellular solution was composed of (in mM) 135 K-Gluconate, 10 HEPES, 4 KCl, 1 EGTA, 0.3 Na-GTP, 4 Mg-ATP, 10 Na2-phosphocreatine (pH: 7.2–7.3, 290–300 mOsm). Recordings were performed at room temperature with constant perfusion of oxygenated neuronal aCSF. Pipette pressure during patch clamp steps was digitally controlled and pipettes were cleaned according to *Kolb et al., 2016*; *Kolb et al., 2019*, as previously described.

## Patch-walking experimental method

The patch-walking algorithm is depicted schematically in *Figure 4*. At the beginning of each patch-walking experiment, cells are first selected by the user. For each slice experiment, we selected 8–10 healthy cells located 20–100 μm below the surface of the tissue. These cells were spread across an area of approximately 200 μm x 200 μm (note the field of view under ×40 magnification is approximately 50 μm). These cell locations with three dimensional coordinates are stored in a cell queue for subsequent patch attempts. From these cell coordinates, the robot computes the distance between the pipettes' respective home positions and each cell.

The assignment of cells to pipettes proceeds as follows. When a pipette is available for a patch clamp attempt, the cell with the shortest distance to the pipette home position is removed from the cell queue and assigned to the pipette. Thus, each pipette initially attempts to patch its closest cell. To probe connections between neurons, both pipettes must form a successful connection. Therefore, in the event that an attempted connection fails, the pipette is retracted and the next closest cell is assigned from the queue. This process is repeated until both pipettes are simultaneously connected to neurons. At this moment, the connections can be tested as described in 'connectivity testing.' A pipette that achieves whole cell configuration is held while patch attempts are made with the other pipette. Unsuccessful patch attempts result in the pipette being cleaned (*Kolb et al., 2019*), assigned a new cell and then a new attempt. For each patch clamp attempt, control of the microscope

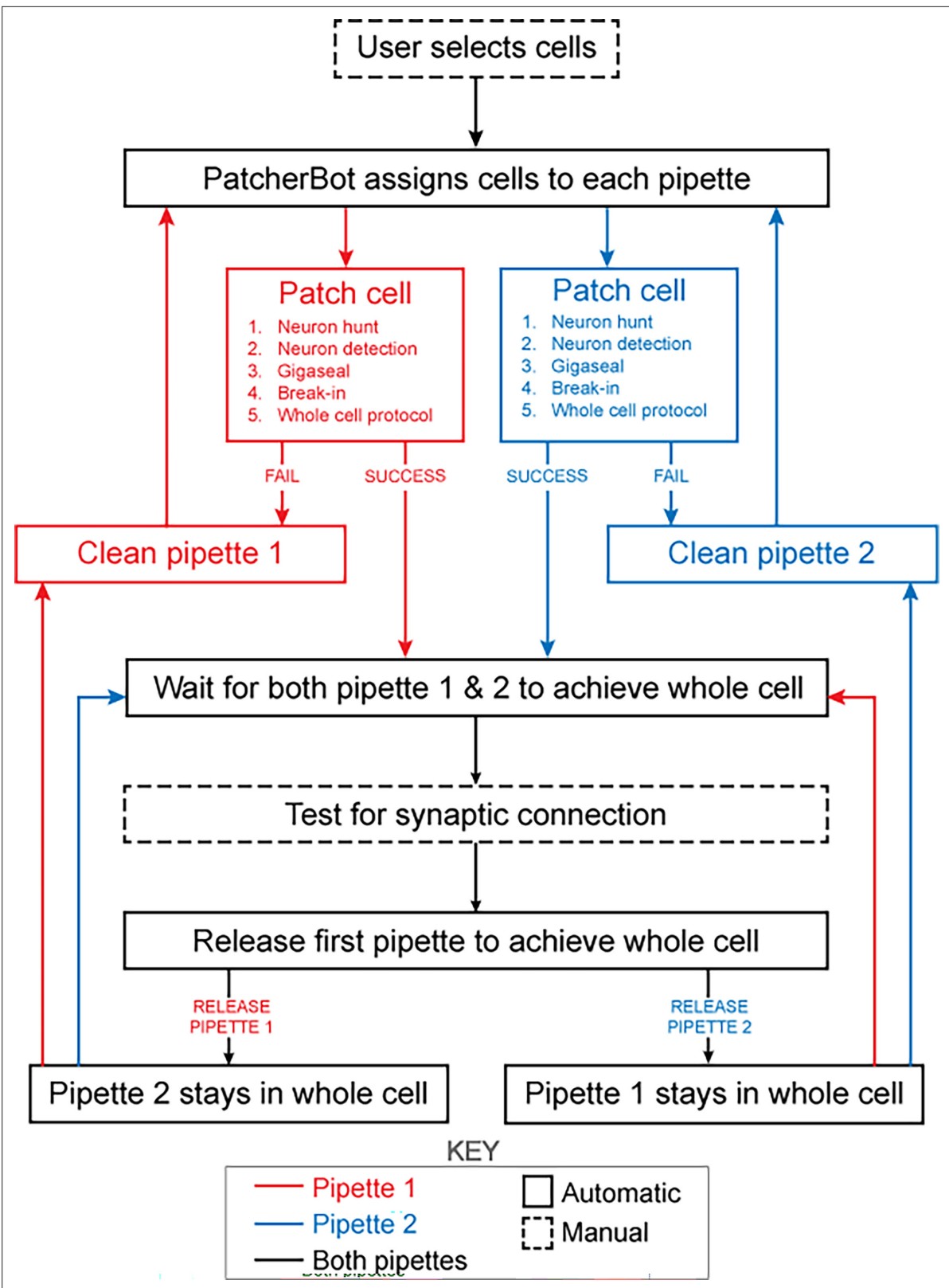

**Figure 4.** Schematic of the patch-walking experimental workflow. The patch-walking process begins with selection of cells by the user. The Patcherbot then assigns cells to each pipette based on their distance to the pipettes' home positions. Each pipette works in parallel, only working independently during steps which require the camera and stage (ie neuron hunting, neuron detection). Once a pipette has achieved these steps successfully, the stage and camera are designated to the other pipette. If the pipette failed the patch attempt, it is cleaned and reused. Once both pipettes achieved whole

*Figure 4 continued on next page*

*Figure 4 continued*

cell configuration, they are tested for synaptic connectivity. In order to ''patch-walk,' the first pipette to achieve whole cell is released to clean and obtain a new whole cell recording.

objective is assigned to the pipette in the 'pipette finding' or 'neuron hunting' phase, until the pipette has successfully completed 'neuron hunting' (when the pipette resistance increases by 0.2 MΩ over 5 descending 0.1 μm steps) (*Kodandaramaiah et al., 2018*). Once both pipettes have established whole cell patch clamp recordings, the connection test is performed. This algorithm then repeated the process until all viable neurons had been patched.

The 'patch cell' step in *Figure 1* includes the following: neuron hunting, neuron detection, giga-sealing, break in, whole cell protocol. As in *Kolb et al., 2019*, and briefly restated here, once the measured resistance reaches 1 GΩ, the algorithm waits (5 s) and proceeds to the break-in state. Break-in is accomplished by short pulses of suction (100–1000 ms, –345 mBar). A break-in is considered successful when the measured resistance drops to under 800 MΩ and the holding current remains low (<–200 pA at –70 mV in slices). The whole cell electrophysiology protocol consists of a voltage clamp protocol where cell parameters (access resistance, membrane resistance, holding current) are measured as well as a current clamp protocol (0 pA for 1 s, –300 to +300 pA step for 1 s, 0 pA for 1 s). Injected current pulses were 3 s pulses from –20 pA to +280 pA in 20 pA steps with a 2 s, –20 pA hyperpolarizing step 500ms prior.

We define a paired recording as a pair of cells which are simultaneously patch clamped using the patch-walking experimental method. We define probed connections as twice the number of paired recordings, because the connections can be bi-directional. We define possible connections as the theoretical upper limit of probed connections given a number of pipettes and number of cells recorded. Probed connections is, practically speaking, less than possible connections since patch clamp yield is 30–80%based on our experience with the PatcherBot, depending on sample preparation and tissue and cell type.

## Recording quality criteria

The access resistance for the neurons in paired recordings were below 40 MΩ, similar to the metric used by *Kolb et al., 2019*, and if the access changed above 50 MΩ, we stopped recording from that neuron. If the seal quality decreased during recording, the cell is excluded from analysis.

## Connectivity testing

We tested for connectivity in a manner similar to that done previously by *Perin et al., 2011* and the Allen Institute for Brain Science (*Campagnola et al., 2022*). To perform connectivity testing between two simultaneously patched neurons, we performed the following procedure. The BNC cables were manually moved from the NI DAQ to the Digitizer to enable Clampex control of the cells (rather than LabView). Two protocols were run in order to test for the two possible directions of connectivity. For each protocol, one pipette sent a stimulus in current clamp mode to elicit five action potentials at 20 Hz while the other pipette holding in voltage clamp recorded post-synaptic currents, held at –70 mV. Following this bi-directional measurement, the BNC cables were replaced manually to resume patch-walking.

## Acknowledgements

This work was funded by the NIH BRAIN Initiative Grant (NEI and NIMH 1-U01-MH106027-01), NIH R01NS102727, NIH Single Cell Grant 1 R01 EY023173, NSF (EHR 0965945 and CISE 1110947), and NIH R01DA029639.

## Additional information

### Competing interests

Ilya Kolb: IK., WMS, and CRF are co-inventors on a patent (US10830758B2) describing pipette cleaning that is licensed by Sensapex. William M Stoy, Craig R Forest: IK, WMS, and CRF are co-inventors on a patent (US10830758B2) describing pipette cleaning that is licensed by Sensapex. Matthew JM Rowan: Reviewing editor, eLife. The other authors declare that no competing interests exist.

## Funding

| Funder | Grant reference number | Author |
|---|---|---|
| National Institutes of Health | R01NS102727 | Mighten C Yip<br>Mercedes M Gonzalez<br>Edward S Boyden<br>Craig R Forest |
| NEI and NIMH | 1-U01-MH106027-01 | Craig R Forest |
| National Institutes of Health Single Cell | 1 R01 EY023173 | Craig R Forest |
| National Science Foundation | EHR 0965945 | Craig R Forest |
| National Science Foundation | CISE 1110947 | Craig R Forest |
| National Institutes of Health | R01DA029639 | Mighten C Yip<br>Edward S Boyden<br>Craig R Forest |

The funders had no role in study design, data collection and interpretation, or the decision to submit the work for publication.

## Author contributions

Mighten C Yip, Conceptualization, Data curation, Software, Formal analysis, Validation, Investigation, Visualization, Methodology, Writing – original draft, Writing – review and editing; Mercedes M Gonzalez, Data curation, Formal analysis, Validation, Investigation, Writing – review and editing; Colby F Lewallen, Conceptualization, Investigation, Methodology, Writing – review and editing; Corey R Landry, Conceptualization, Methodology, Writing – review and editing; Ilya Kolb, Conceptualization, Writing – review and editing; Bo Yang, Writing – review and editing; William M Stoy, Conceptualization; Ming-fai Fong, Formal analysis, Supervision, Writing – review and editing; Matthew JM Rowan, Supervision, Writing – review and editing; Edward S Boyden, Supervision, Funding acquisition, Writing – review and editing; Craig R Forest, Conceptualization, Supervision, Funding acquisition, Writing – original draft, Project administration, Writing – review and editing

## Author ORCIDs

Mighten C Yip https://orcid.org/0000-0002-8463-0311
Ming-fai Fong https://orcid.org/0000-0002-2336-4531
Edward S Boyden https://orcid.org/0000-0002-0419-3351
Craig R Forest https://orcid.org/0000-0001-5343-1769

## Ethics

All animal procedures were in accordance with the US National Institutes of Health Guide for the Care and Use of Laboratory Animals and were approved by the Institutional Animal Care and Use Committee at the Georgia Institute of Technology (A100359).

Reviewer #1 (Public review): https://doi.org/10.7554/eLife.97399.3.sa1
Reviewer #2 (Public review): https://doi.org/10.7554/eLife.97399.3.sa2
Reviewer #3 (Public review): https://doi.org/10.7554/eLife.97399.3.sa3
Author response https://doi.org/10.7554/eLife.97399.3.sa4

## Additional files

### Supplementary files
• MDAR checklist

### Data availability
Custom LabVIEW code for controlling the patcherBot is publicly available on autopatcher.org. Pipette cleaning software also available on Github (https://github.com/mightenyip/Pipette-Cleaning-Software, copy archived at *mightenyip, 2024*) under a MIT License. Source data needed to recreate the primary results shown in Figures and electrophysiological recordings are available in Dryad.

The following dataset was generated:

| Author(s) | Year | Dataset title | Dataset URL | Database and Identifier |
|---|---|---|---|---|
| Yip MC | 2024 | Patch-walking electrophysiology recordings | https://doi.org/10.5061/dryad.x69p8cztq | Dryad Digital Repository, 10.5061/dryad.x69p8cztq |

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
