## [Editor Report · eLife Assessment]

This technical study presents a novel sampling strategy for detecting synaptic coupling between neurons from dual pipette patch-clamp recordings in acute slices of mammalian brain tissue in vitro. The authors present **solid** evidence that this strategy, which incorporates automated patch clamp electrode positioning and cleaning for reuse with strategic neuron targeting, has the potential to substantially improve the efficiency of neuronal sampling with paired recordings. This technique and the extensions discussed will be **useful** for neuroscientists wanting to apply or already conducting automated multi-pipette patch clamp recording electrophysiology experiments in vitro for neuron connectivity analyses.

---

## [Referee Report · Reviewer #1 (Public review)]

Summary:

In this technical paper, the authors introduce an important variation on the fully automated multi-electrode patch-clamp recording technique for probing synaptic connections that they term "patch-walking". The patch-walking approach involves coordinated pipette route-planning and automated pipette cleaning procedures for pipette reuse to improve recording throughput efficiency, which the authors argue can theoretically yield almost twice the number of connections to be probed by paired recordings on a multi-patch electrophysiology setup for a given number of cells compared to conventional manual patch-clamping approaches used in brain slices in vitro. The authors show convincing results from recordings in mouse in vitro cortical slices, demonstrating the efficient recording of dozens of paired neurons with a dual patch pipette configuration for paired recordings and detection of synaptic connections. This approach will be of interest and valuable to neuroscientists conducting automated multi-patch in vitro electrophysiology experiments and seeking to increase efficiency of neuron connectivity detection while avoiding the more complex recording configurations (e.g., 8 pipette multi-patch recording configurations) used by several laboratories that are not readily implementable by most of the neuroscience community.

Strengths:

(1) The authors introduce the theory and methods and show experimental results for a fully automated electrophysiology dual patch-clamp recording approach with a coordinated patch-clamp pipette route-planning and automated pipette cleaning procedures to "patch-walk" across an in vitro brain slice.

(2) The patch-walking approach offers throughput efficiency improvements over manual patch clamp recording approaches, especially for investigators looking to utilize paired patch electrode recordings in electrophysiology experiments in vitro.

(3) Experimental results are presented from in vitro mouse cortical slices demonstrating the efficiency of recording dozens of paired neurons with a two-patch pipette configuration for paired recordings and detection of synaptic connections, demonstrating the feasibility and efficiency of the patch-walking approach.

(4) The authors suggest extensions of their technique while keeping the number of recording pipettes employed and recording rig complexity low, which are important practical technical considerations for investigators wanting to avoid the more complex recording configurations (e.g., 8-10 pipette multi-patch recording configurations) used by several laboratories that are not readily implementable by most of the neuroscience community.

---

## [Referee Report · Reviewer #2 (Public review)]

Summary:

In this study, the authors aim to combine automated whole-cell patch clamp recording simultaneously from multiple cells. Using a 2-electrode approach, they are able to sample as many cells (and connections) from one slice, as would be achieved with a more technically demanding and materially expensive 8-electrode patch clamp system. They provide data to show that this approach is able to successfully record from 52% of attempted cells, which was able to detect 3 pairs in 71 screened neurons. The authors state that this is a step forward in our ability to record from randomly connected ensembles of neurons.

Strengths:

The conceptual approach of recording multiple partner cells from another in a step wise manner indeed increases the number of tested connections. An approach that is widely applicable to both automated and manual approaches. Such a method could be adopted for many connectivity studies using dual recording electrodes.

The implementation of automated robotic whole-cell patch-clamp techniques from multiple cells simultaneously is a useful addition to the multiple techniques available to ex vivo slice electrophysiologists.

The approach using 2 electrodes, which are washed between cells is economically favourable, as this reduces equipment costs for recording multiple cells, and limits the wastage of capillary glass that would otherwise be used once.

Weaknesses:

(1) Based on the revised manuscript - a discussion of the implementation of this approach to manual methods is still lacking,

(2) A comparison of measurements shown in Figure 2 to other methods has not been addressed adequately.

(3) The morphological identification of neurons is understandably outside the remit of this project - but should be discussed and/or addressed. It was not suggested to perform detailed anatomical analysis - but to highlight the importance of this, and it should still be discussed

(4) The revised manuscript does not clearly state which cells were included in the analysis as far as I can see - and indeed cells with Access Resistance >40 MOhm appear to still be included in the data.

---

## [Referee Report · Reviewer #3 (Public review)]

Summary:

In this manuscript, Yip and colleagues incorporated the pipette cleaning technique into their existing dual-patch robotic system, "the PatcherBot", to allow sequential patching of more cells for synaptic connection detection in living brain slices. During dual-patching, instead of retracting all two electrodes after each recording attempt, the system cleaned just one of the electrodes and reused it to obtain another recording while maintaining the other. With one new patch clamp recording attempt, new connections can be probed. By placing one pipette in front of the other in this way, one can "walk" across the tissue, termed "patch-walking." This application could allow for probing additional neurons to test the connectivity using the same pipette in the same preparation.

Strengths:

Compared to regular dual-patch recordings, this new approach could allow for probing more possible connections in brain slices with dual-patch recordings, thus having the potential to improve the efficiency of identifying synaptic connections

Weaknesses:

While this new approach offers the potential to increase efficiency, it has several limitations that could curtail its widespread use.

Loss of Morphological Information: Unlike traditional multi-patch recording, this approach likely loses all detailed morphology of each recorded neuron. This loss is significant because morphology can be crucial for cell type verification and understanding connectivity patterns by morphological cell type.

Spatial Restrictions: The robotic system appears primarily suited to probing connections between neurons with greater spatial separation (~100µm ISD). This means it may not reliably detect connections between neurons in close proximity, a potential drawback given that the connectivity is much higher between spatially close neurons. This limitation could help explain the low connectivity rate (5%) reported in the study.

Limited Applicability: While the approach might be valuable in specific research contexts, its overall applicability seems limited. It's important to consider scenarios where the trade-off between efficiency and specific questions that are asked.

Scalability Challenges: Scaling this method beyond a two-pipette setup may be difficult. Additional pipettes would introduce significant technical and logistical complexities.

---

## [Author Response]

The following is the authors’ response to the original reviews.

We thank the reviewers and editors for insightful feedback on how we could improve the manuscript. We have revised the manuscript and addressed the points raised.

Regarding the technical issues raised about the quality of patch clamp recordings (Reviewer 2), we acknowledge that the upper limit of the access resistance cutoff should be lower and that the accepted change should be 10-20%. To this end, we have revised the manuscript to more accurately detail the quality metrics used. The access resistance for the neurons in paired recordings were below 40 MΩ (similar to the metric used by Kolb et al. 2019), and if the access changed above 50 MΩ, we stopped recording from that neuron. Furthermore, the inclusion of neurons in the histogram with access resistance above 50 MΩ was to highlight the total number of neurons patched but not necessarily used in paired recordings. As this was done with an automated robotic system, the neurons would still undergo an initial voltage clamp and current clamp protocol before the pipette would release the neuron and patch another cell. To the point of Reviewer 2, this patch-walk protocol could also be alternatively implemented using manual recording approaches and this point has been included in the revised manuscript.

Regarding the spatial restrictions (Reviewer 3), we agree that the average intersomatic distance is higher than ideal. This was likely due to failed patch attempts; for instance, if one pipette successfully achieved whole cell, and the other pipette had several sequential failed patch attempts, the intersomatic distance (ISD) would increase with each failed attempt due to the user selected index of cells. Ideally, the pipettes would be walking across a slice with low ISD if the whole-cell success rate was closer to 100%. To overcome this challenge in future work, automated cell identification and tracking could enable the path planning to be continuously updated after each patch attempt. Given the whole-cell success rate efficiency for a given electrophysiologist, we believe that the automated robot could be improved in later versions to include routeplanning algorithms to minimize the distance between neurons. Alternatively, this patch-walk system could also be integrated to improve connectivity yields for manual recording approaches as well.

For the point raised about morphological identification, we believe that while important, morphological identification is out of the scope for this project. Future work will include neuronal reconstruction. Regarding the other points, we will amend the manuscript to highlight other key metrics such as maximum time we could hold a neuron under the whole-cell configuration. Additionally, we agree with Reviewer 3 that some of the current language may cause confusion, and we will amend it accordingly.

To all the reviewers, thank you for your time, understanding, and the opportunity to improve our manuscript.